# A comparative study of choroidal thickness and pigment epithelial detachment in acute and chronic central serous chorioretinopathy in Nepalese patients

**Subash Bhatta**[1]◉*, **Nayana Pant**[1]◉, **Suresh Raj Pant**[2]

**1** Drishti Eye Hospital, Jhapa, Nepal, **2** Geta Eye Hospital, Kailali, Nepal

◉ These authors contributed equally to this work.
* subashbhatta@gmail.com

## Abstract

### Background

To study the significance and correlation of choroidal and retinal pigment epithelial changes with disease activity in Central Serous Chorioretinopathy (CSCR) eyes.

### Methods

This was a retrospective analysis of clinical records and optical coherence tomography (OCT) images of CSCR cases presenting to a tertiary eye hospital in Nepal between October 2021 to November 2022. The study included 145 CSCR eyes from 132 cases compared with 290 eyes of 145 age- and sex-matched healthy volunteers. Chi square test, Paired T-test and Independent sample t-test were used for statistical analysis.

### Results

Average subfoveal choroidal thickness (SFCT) of the CSCR eyes (453.13 um) and the fellow eyes (403.44 um) was significantly greater (p < 0.001) than that of the control group (372.29um). Notably, the eyes affected by CSCR also had significantly greater SFCT than fellow eyes (p < 0.01). No significant correlation was observed between subfoveal choroidal thickness (SFCT) and either central subfield macular thickness (CST) (p = 0.559) or pigment epithelial detachment (PED) (p = 0.145). Chronic CSCR eyes showed a significant association with PED (Chi-square test, $p < 0.013$), and a trend toward reduced CST and SFCT compared to acute eyes, as indicated by the independent t-tests ($p = 0.04$ and $p = 0.023$, respectively). Flat and irregular PEDs were more common in chronic CSCR eyes compared to acute CSCR eyes (p = 0.027).

**Data availability statement:** All relevant data are within the paper and its Supporting information files.

**Funding:** The author(s) received no specific funding for this work.

**Competing interests:** The authors have declared that no competing interests exist.

## Conclusions

Increased SFCT and PED are significant pathophysiological markers in CSCR, exhibiting distinct variations between acute and chronic forms. However, the lack of a direct correlation of SFCT with CST and PED underscores the limitation of relying solely on SFCT to fully characterize choroidal changes in CSCR. Further exploration of additional OCT biomarkers may offer deeper insights into the complex pathophysiology of these changes, paving the way for enhanced understanding and more targeted therapeutic strategies.

## Introduction

While most cases of Central serous chorioretinopathy (CSCR) are self-limiting, some cases develop chronic course, exhibiting RPE degeneration, foveal atrophy or choroidal neovascular membrane (CNVM), resulting in vision loss [1]. Optical coherence tomography (OCT ) has significantly enhanced our understanding of disease pathology aiding in diagnosis, classification and treatment of the disease [2]. Improved choroidal imaging using swept source OCT and enhanced depth imaging (EDI) technique in spectral domain OCT have further illuminated the role of choroid in CSCR pathophysiology. It is now widely accepted that alterations in choroidal circulation play a pivotal role in the disease process [3]. It is presumed that hyperpermeability of choroidal vessels in CSCR leads to increased tissue hydrostatic pressure and the formation of PEDs [4]. These PEDs present as dome-shaped or flat and irregular lesions in OCT. The most common used measure of choroidal circulatory abnormality in OCT imaging is subfoveal choroidal thickness (SFCT). Although choroidal thickness and PED seem to be causally linked in disease process, correlation between these parameters remain underrepresented in scientific research. The significance of PED morphology in pathophysiology of CSCR is also not clear.

We thus conducted this study to understand the significance of SFCT and PED in CSCR in relation to disease activity. As SFCT values can differ among various populations and there was no data on the average SFCT values for our local population, we also measured the SFCT of age- and sex-matched healthy adults to compare it with the choroidal thickness in CSCR eyes.

## Methods

This was a retrospective analysis of clinical records and OCT images of all CSCR cases presenting to a tertiary eye hospital in Nepal between first of October 2021 to end of November 2022. The research study adhered to the principles outlined in the Declaration of Helsinki. Prior to participation, written informed consent was obtained from all subjects. Ethical approval for the study was granted by the Nepal Netra Jyoti Sangh's ethical review committee.

Cases with significant ocular co-morbidities were excluded from the study. All the CSCR patients underwent clinical examination by ophthalmologists followed by OCT examination using a standard protocol. The study group

initially consisted of 161 CSCR eyes from 147 cases. Sixteen eyes were excluded as OCT images were considered ungradable. After exclusion of these eyes, 145 CSCR eyes from 132 cases were available for final analysis. Additionally, we enrolled 145 age- and sex-matched healthy volunteers with no known eye diseases, and no prior corticosteroid use as controls. Only eyes with a normal, fundus and OCT findings, and spherical equivalent refractive error between +1.00 D and −1.00 D were included in the control group. Control eyes were included to establish baseline subfoveal choroidal thickness within the study population, enabling comparison and evaluation of SFCT alterations in eyes affected by CSCR.

We divided the study eyes with CSCR into acute and chronic groups. In a 2015 review, Daruich et al defined acute CSCR cases as eyes with the first episode of disease and subretinal detachment (SRD) lasting less than 4 months [5]. Persistent CSCR referred to eyes with SRD lasting more than 4 months. Recurrent CSCR was used to describe new SRD occurring after a period of disease inactivity in known CSCR cases. In our study, we adopted the same criteria for defining acute CSCR. The term chronic CSCR was used to encompass cases of persistent and recurrent presentations, following the classification outlined by Daruich et al. [5].

## OCT protocol

A 9 by 9 mm 24-line volume scan was done using the Spectralis Spectral Domain (SD) OCT system (Heidelberg Engineering, Heidelberg, Germany). An automatic real-time (ART) averaging of 100 frames was applied to the sub foveal section and real time eye-tracking was used in order to obtain quality images for evaluating choroidal changes including choroidal thickness. Single 9 mm horizontal EDI OCT line scan passing through the centre of the fovea was used for analysis of sub foveal choroidal thickness.

Inner and outer choroidal borders were identified, and sub foveal choroidal thickness (SFCT), defined as the distance between these borders, was measured with callipers inbuilt in the software. Inner choroidal border was defined as the outer edge of the hyperreflective line corresponding to the retinal pigment epithelium and Bruch's membrane. When Choroidal scleral interface (CSI) was visible as a hyper-reflective band outer to the vascular-like structure of the choroid, the outer limit of the CSI was used as the outer choroidal boundary. Inner limit of the suprachoroidal space (SCS), defined as a hypo-reflective band posterior to the choroid, was used as the boundary if CSI was not visible. Perceptible imaginary line signifying the outer limit of the large choroidal vessels was used in case when no clear CSI or SCS was visible. Scans were considered to be ungradable if there was no identifiable posterior boundary of the choroid or portion of choroid was missing on the scan.

The EDI OCT images for both eyes were evaluated by two masked expert observers. If any of the two observers deemed an image to be of poor quality to grade or the observers had more than 5% difference in measurement of choroidal thickness, the image was considered ungradable and the case or the control was excluded from the study. Average values between the two grades were used for the final analysis.

All OCT slices within 9 by 9 mm 24-line volume scan were used to analyse and define the morphology of PEDs in CSCR cases. Flat PEDs (sometimes, also referred as flat and wavy or flat and irregular PEDs in literature) were segregated from dome shaped PEDs as literature suggest that presence of flat PEDs may indicate presence of neovascular changes under RPE, pointing to a different pathophysiological process [6,7]. Double layer sign is a common term used for OCT finding to describe this separation of the RPE layer from the inner layer of the Bruch's membrane in form of flat and irregular PEDs [8].

Central subfield macular thickness (CST), defined as the average retinal thickness within a 1 mm diameter circle centred on the fovea, was utilized to evaluate central retinal thickening resulting from subretinal fluid accumulation in CSCR eyes. The value was automatically generated by the OCT software.

A total sample size of 105 was calculated by considering a margin of error of 5%, a confidence level of 95% and power of 0.8. Chi square test was used to analyse categorical variables like age group and PED morphology. Paired T-test and

independent sample t-test were used to analyse between continuous variables like SFCT and CST between groups. All statistical analyses were performed in SPSS-20 software. Tests were considered significant at p < 0.05.

We pre-specified three primary hypotheses to focus on the study's main aims: (1) difference in SFCT between eyes with and without PED; (2) association of PED presence with disease activity (acute vs chronic CSCR); and (3) difference in SFCT between acute and chronic CSCR eyes. Bonferroni correction was applied to these three primary tests to control family-wise error (α = 0.05/3 = 0.0167). Additional analyses, including PED morphology and SFCT–CST correlation, were considered exploratory and were interpreted without multiplicity correction.

## Results

The median age of the study group was 37 ± 6.96 years. CSCR was more frequently seen in the age group of 31–50 years (n = 110, 83.3%) which is statistically significant according to the chi-square test (p < 0.05) (Table 1). There were significantly more male cases (n = 99, 75%) compared to the females (p < 0.05).

The difference in choroidal thickness between the right eye and left eye in the control group was not significant (paired sample t-test = 0.308). Independent sample t-tests showed no significant difference in choroidal thickness between male and female CSCR cases (p = 0.362). Both affected eyes and fellow eyes in CSCR cases exhibited greater SFCT compared to eyes in the control group (p < 0.05). Notably, the eyes affected by CSCR also had significantly greater SFCT than fellow eyes (p < 0.05). (Table 2).

The Chi-square test revealed that PED is significantly associated with affected eyes in CSCR patients (p < 0.05). (Table 3). PED also had significant correlation with CST (p < 0.05) in these eyes (Table 4). However, SFCT and PED did not show significant (p = 0.559) correlation with each other. There was also no significant correlation between SFCT with CST (p = 0.145).

PEDs were more commonly present in chronic CSCR eyes compared to acute CSCR eyes (Chi-square test, p < 0.013) and this association remained significant after Bonferroni correction. (Table 5). Although there was no significant

**Table 1. Demographic distribution of CSCR cases and controls.**

| Age group | Case | | | Control | | |
|---|---|---|---|---|---|---|
| | Male | Female | Total | Male | Female | Total |
| 21-30 | 13 | 3 | 16 | 13 | 3 | 16 |
| 31-40 | 48 | 18 | 66 | 56 | 20 | 76 |
| 41-50 | 34 | 10 | 44 | 37 | 10 | 47 |
| 51-60 | 4 | 2 | 6 | 4 | 2 | 6 |
| Total | 99 | 33 | 132 | 110 | 35 | 145 |
| % of total | 75% | 25% | | 76% | 24% | |

**Table 2. Mean sub foveal choroidal thickness (SFCT) of study and control eyes.**

| Gender | Male | Female | Total |
|---|---|---|---|
| Number of all CSCR eyes | 110 | 35 | 145 |
| SFCT in CSCR eyes (micrometre) | 447.59 | 470.54 | 453.13 |
| Number of fellow eyes without CSCR | 88 | 31 | 119 |
| SFCT in fellow eyes (micrometre) | 401.68 | 408.45 | 403.44 |
| SFCT in Control RE | 376.42 | 369.14 | 374.66 |
| SFCT in Control LE | 370.95 | 366.68 | 369.92 |
| p-value* | <0.05 | | |

*p-value: differences in SFCT between CSCR eyes with fellow eyes and control eyes.

difference in presence of isolated flat PEDs between acute and chronic CSCR eyes, we found that overall flat PEDs (present solitarily or mixed with dome shaped PEDs) were significantly more common in chronic CSCR eyes (p = 0.027).

Central subfield macular thickness was greater in acute CSCR eyes compared to chronic cases (Independent T-test, p = 0.04). Subfoveal choroidal thickness was greater in acute CSCR eyes (464.7 1 μm) compared to chronic CSCR eyes (419.3 μm), with a mean difference of 45.4 μm (Independent t test, p = 0.023). However, this difference did not remain significant after Bonferroni correction (adjusted α = 0.0167).

## Discussion

This is the largest series of CSCR cases reported from Nepal. Among the 132 CSCR cases included in the study, three fourths were male patients and more than 80% of patients were within the age group of 31–50 years. These findings align with demographic results observed in studies conducted across various geographical areas and populations [5,9].

**Table 3. Pigment epithelial detachment (PED) among affected and fellow eyes of CSCR cases.**

| Eyes with CSCR | Eyes with CSCR, 145 (100%) | Fellow eyes without CSCR (Unilateral cases), 119 (100%) | Chi-square test |
|---|---|---|---|
| No PED | 52 (35.9%) | 90 (75.6%) | P=<0.05 |
| Dome shaped PED | 37 (25.5%) | 17 (14.3%) | |
| Flat PED | 38 (26.2%) | 9 (7.6%) | |
| Mixed (Flat and Dome shaped) PED | 18 (12.4%) | 3 (2.5%) | |

**Table 4. Association between PED with CST and SFCT in CSCR eyes.**

| PED | Frequency | CST (micrometer) | SFCT (micrometer) |
|---|---|---|---|
| Absent | 52 | 261.46 | 459.98 |
| Dome shaped PED | 37 | 355.86 | 443.84 |
| Flat PED | 38 | 321.57 | 457.47 |
| Mixed (Flat and Dome shaped) PED | 18 | 295.27 | 443.28 |
| | | *P < 0.05 | **P = 0.559 |

SFCT: Subfoveal choroidal thickness, CST: Central subfield macular thickness.

*Correlation of PED with CST, **Correlation of PED with SFCT.

**Table 5. Association of SFCT, PED morphology, and CST with disease activity in CSCR eyes.**

| | Acute CSCR eyes | Chronic CSCR eyes | All eyes with CSCR |
|---|---|---|---|
| Total | 108 | 37 | 145 |
| All PED | 63 (58.3%) | 30 (81%) | 93 (64.1%) |
| Flat PED | 28 (25.9%) | 10 (27.0%) | 38 (26.2%) |
| Dome shaped PED | 23 (21.3%) | 14 (37.8%) | 37 (25.5%) |
| Mixed (Flat and Dome shaped) PED | 12 (11.1%) | 6 (16.2%) | 18 (12.4%) |
| SFCT (micrometer) | 464.71 | 419.32 | 453.13 |
| CST (micrometer) | 324.99 | 248.62 | 305.50 |

SFCT: Subfoveal choroidal thickness, CST: Central subfield macular thickness.

CST compared between acute and chronic CSCR eyes (Independent t-test, p = 0.04).

SFCT compared between acute and chronic CSCR eyes (Independent t-test, p = 0.023).

The average SFCT for the healthy age and sex matched control population (median age: 36±7.45 years) was 372 micrometer (μm) in our study. We observed that there was no significant difference in SFCT values between male and female subjects, as well as between the right and left eyes in the control group. Notably, these findings also align with another report from a different part of the country [10]. Compared to our study, they reported a slightly lesser mean SFCT of 353.24±65.63 μm in a healthy Nepalese population. The higher average SFCT values in our study may be attributed to the younger age demographics of our subjects. Notably, SFCT values also vary across different populations. For instance, the Beijing Eye study found thinner SFCT among the normal population, with a mean of 253.8±107.4 μm using EDI OCT [11]. A study of SFCT in the English population, using the same machine and method, showed a mean SFCT of 332 μm (range: 142–563 μm) [12]. The differences in SFCT values may reflect anatomical differences in choroidal thickness in different populations. This makes it challenging to interpret SFCT measurements without standard data for the given population. However, based on different cohorts, Lehman et al proposed 395 micron as cut off value to consider thickened choroid in their study [13].

In our study the mean SFCT of CSCR eyes was 453 μm and the unaffected fellow eyes had an average SFCT of 403μm. The SFCT was significantly greater in CSCR eyes compared to the fellow eyes and the control eyes, as observed in different other studies [14–19]. These observations have supported the theory that CSCR falls into a group of pachy-choroid spectrum diseases comprising other pathologies like pachychoroid pigment epitheliopathy, pachychoroid neo-vasculopathy and polypoidal choroidal vasculopathy [20,21]. The fellow eyes also had significantly thicker SFCT than in normal subjects. This finding supports the argument that the underlying pathology in CSCR is essentially a bilateral process with fellow eye being at risk for CSCR or pigment epitheliopathy.

It is suggested that an abnormal increase in choroidal thickness leads to choroidal congestion and prolonged second-ary backpressure resulting from choroidal congestion can cause structural damage to the retinal pigment epithelium and Bruch's membrane [17]. These changes are reflected in form of RPE irregularities, pigment epithelial detachment and subretinal fluid accumulation.

In our study, 64.1% of CSCR cases exhibited some form of PED in the affected eye which was significantly higher than in the fellow eyes where PED was present in 24.4% of cases. Among CSCR cases, a larger proportion of chronic CSCR eyes (81%, n = 30) had PEDs compared to acute CSCR eyes (58.3%, n = 63). Esroz et al. reported PEDs in 80.7% of their cases and highlighted a significant association between PEDs and chronic CSCR [9]. In a 2020 study by Argyrios et al., PED was present in 90% of cases, which exceeded the prevalence observed in our study [22]. Their follow-up data supported the association between PED and the development of chronic and recurrent CSCR, consistent with our findings highlighting a very strong association of PED with chronic CSCR eyes compared to acute CSCR eyes.

Some studies have reported the presence of fibrovascular components in flat and irregular PEDs [23,24]. Bosquet et al. found a fibrovascular component in as many as one-third of their patients with flat and irregular PEDs. In their study, only flat PEDs with hyperreflective contents were associated with neovasculopathy. In our study, we did not assess the content of all flat PEDs with angiography. However, a significant number of these flat PEDs were observed even in the early course of the disease, and many were also present concurrently with dome-shaped PEDs. Hence, we anticipate that a substantial proportion of these PEDs may not be associated with neovascular changes. Given the reports of fibrovas-cular components in these flat PEDs, it may be logical to evaluate them using optical coherence tomography angiography (OCTA) or conventional angiography, especially if they exhibit hyperreflective content and the disease is persistent or recurrent. Our study also demonstrated a significantly higher number of flat PEDs in chronic and recurrent cases, similar to findings reported in some other studies [9,22].

Although PED and subretinal detachments in CSCR are considered to be causally linked to choroidal vascular abnor-malities, our study found that SFCT couldn't be directly correlated to PED formations or increased CST. SFCT is just the measurement of choroidal thickness at one point, and may not represent the overall choroidal vascular problem. Although it is the most commonly used OCT based indicator for choroidal vascular changes in clinical practice, measurement of

subfoveal choroidal thickness still has some inherent challenges. Most of the machines don't have default standardized software tools to measure choroidal thickness and sub foveal choroidal thickness measurements are subject to inter-observer or intra-observer variations. The ability to assess the outer boundary of the choroid depends upon the thickness of the retina and choroid. Specifically, thicker retina and choroid make the demarcation line between the choroid and sclera less prominent, as light or laser has to pass through greater depths of tissues to reach this structure [12]. This difficulty in accurately measuring SFCT values is particularly relevant in cases of CSCR eyes as the retinal thickness can be significantly increased due to the subretinal fluid. Approximately 10% of CSCR eyes did not yield reliable SFCT measurements and were excluded from our study. However, advancements in OCT technologies, which offer better choroidal penetration and automated choroidal thickness measurements, hold promise for enhancing SFCT assessment.

Another limitation with SFCT measurement in pachychoroid spectrum disease is its inability to distinguish between vascular and stromal components [25]. A study comparing choriocapillaris features between three groups namely; (a) healthy eyes with normal choroidal thickness (b) healthy eyes with thickened choroid, and (c) CSCR eyes reported no significant differences in choriocapillaris parameters between healthy eyes with and without a thick choroid. However, in eyes with CSCR, the choriocapillaris showed reduced capillary density, with vessels appearing longer and wider [26]. This finding suggests that CSCR may be associated with choriocapillaris abnormalities independent of choroidal thickness, and that a thick choroid does not always equate to pathology [27]. Emerging biomarkers like choroidal vascularity index (CVI) may serve as a more reliable biomarker than subfoveal choroidal thickness in elucidating the pathophysiology of CSCR as it accounts for both vascular and stromal components of the choroid. Advancements in OCT angiography technology have further enhanced the ability to visualize and quantify choroidal vascular changes, enabling more precise assessment of CVI and its correlation with disease activity [28]. Recent studies have also highlighted the association of scleral thickness in pathogenesis of CSCR, using swept source anterior segment OCT to measure anterior scleral thickness under extra-ocular muscle insertions [29,30]. Additionally, vortex vein congestions are also being studied as potential factors in pathogenesis of CSCR, opening avenues to further modalities and biomarkers to understand the disease [30].

As this was a cross sectional retrospective study with limited sample size, we had limitations in assessing disease activity and progression. Additionally, lack of fluorescein angiography hindered our ability to fully comprehend the pathophysiology of flat and irregular PEDs within our sample. Although the definitions for acute and chronic CSCR used in this study are based on previous literature [5], the definitions are not universally accepted. An effort was made to establish a standardized classification system by the Central Serous Chorioretinopathy International Group; however, it has yet to be validated for universal adoption [31]. The absence of consensus hinders the development of standardized research parameters in CSCR, making it difficult to synthesize consistent and cohesive insights from both current and past studies.

## Conclusion

This study provides average SFCT values for a subset of Nepalese population and highlights the importance of SFCT and PED in understanding disease progression and prognosis. Compared to acute CSCR, chronic cases showed a significantly higher prevalence of PED—particularly flat and irregular types—while acute CSCR eyes showed trend towards greater central subfield thickness and subfoveal choroidal thickness. Although being an significant finding in CSCR eyes, a lack of direct correlation of subfoveal choroidal thickness with PED and central subfield macular thickness in our analysis underscores it's limitations in fully capturing the choroidal pathophysiology affecting the subsequent changes in pigment epithelium and retina. We recommend exploring additional OCT biomarkers for choroidal changes to enhance our understanding of the underlying pathophysiology and prognostic factors in CSCR.

## Supporting information

**S1 Dataset.** *datasetCSCReyes.csv.* Raw data for CSCR eyes group used in analysis.
(CSV)

**S2 Dataset.** *datasetcontroleyes.csv*. Raw data for control eyes used in comparative analysis. (CSV)

**S3 Document.** *Metadata CSCR.docx*. Metadata definitions and descriptions for CSCR eyes dataset variables. (DOCX)

## Acknowledgments

We acknowledge all the staffs of retina department in Geta Eye Hospital for supporting us in conduct this study.

## Author contributions

**Conceptualization:** Subash Bhatta, Nayana Pant.

**Data curation:** Subash Bhatta.

**Formal analysis:** Subash Bhatta, Nayana Pant, Suresh Raj Pant.

**Investigation:** Subash Bhatta, Nayana Pant.

**Methodology:** Subash Bhatta, Nayana Pant.

**Project administration:** Subash Bhatta, Nayana Pant.

**Resources:** Subash Bhatta, Nayana Pant.

**Software:** Subash Bhatta, Nayana Pant.

**Supervision:** Subash Bhatta, Nayana Pant.

**Validation:** Subash Bhatta, Nayana Pant, Suresh Raj Pant.

**Visualization:** Subash Bhatta, Nayana Pant.

**Writing – original draft:** Subash Bhatta, Nayana Pant, Suresh Raj Pant.

**Writing – review & editing:** Subash Bhatta, Nayana Pant, Suresh Raj Pant.

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
