## [Decision Letter · Decision Letter 0]

17 Sep 2025

Comparative analysis of choroidal thickness and pigment epithelial detachment in Acute and Chronic Central serous chorioretinopathy

PLOS ONE

Dear Dr. Bhatta,

Thank you for submitting your manuscript to PLOS ONE. After careful consideration, we feel that it has merit but does not fully meet PLOS ONE’s publication criteria as it currently stands. Therefore, we invite you to submit a revised version of the manuscript that addresses the points raised during the review process.

**ACADEMIC EDITOR: **

1. Please include this is a study of Nepalese patients in the title.

2. The abbreviations and terminology related to subfoveal choroidal thickness, chronic serous chorioretinopathy and subfield choroidal thickness are not carefully defined and appear to be used interchangeably in places. Please review and please be consistent throughout. Thank you

3. The control patients need to be explained clearly. Also in Table 5, there is a column with non-affected fellow eyes? What are the controls specifically used for the study please.

4. The distinction between acute and chronic/persistent (same thing?) disease is not established clearly in the Methods, nor is there comparative data, as noted in the title. Please clarify or change the title to reflect the exact purpose of the study. Thank you.

5. For any measurements in Tables (for example, Table 5), the measurement units (microns) must be included and the number of decimal places biologically meaningful (not to 3 decimal places for example). Any abbreviations used in tables should also be fully explained so that the reader can follow the information easily. Thank you.

6. Table 4 is described as showing associations between the different pathologies, however, no statistics are reported (as far as can tell) and this seems to be a summary table of numbers of patient eyes for the different conditions. Please either anaylse for associations as noted or remove this table. The same comments apply to Table 5, which also notes associations in the title but does not provide any analysis. Thank you.

7. The observations regarding PED, subfoveal choroidal thickness as biomarkers for CSCR (acute or chronic?) are reported previously in literature, and the conclusions around this finding and need for more choroidal biomarkers to be examined requires a more defined and focused statement, supported by the observations. Thank you.

8. The literature cited is limited and not current for this area of ophthalmology, noting the most recent reference cited appears to be 2023. Much has been published in this area related to pathophysiology, classification of pachychoroid spectrum diseases and management since 2023. For example, a review from Khan and Lotery, Ann Rev Vis Sci 2024 doi: 10.1146/annurev-vision-102122-102907, is but one example. Am updated and thorough literature review is required please.

9. With regard limitations, please note the limited (around 140) patient numbers, and be careful using the term 'population' in referring to the study group, noting sample size. Thank you. 

We look forward to receiving your revised manuscript.

Kind regards,

Michele Madigan

Academic Editor

PLOS ONE

2. Please amend either the abstract on the online submission form (via Edit Submission) or the abstract in the manuscript so that they are identical.

Additional Editor Comments:

Reviewer 1

This study examines the relationship of subfoveal choroidal thickness (SFCT) and pigment epithelial detachment (PED) morphology with disease activity in CSCR, using OCT data from a Nepalese cohort. The manuscript provides region-specific normative SFCT values, which are lacking in current literature, and adds to the understanding of pachychoroid spectrum disorders. The findings are clinically relevant, particularly in differentiating acute from chronic disease, and in identifying biomarkers of progression. However, there are view concerns that could impact the clarity of the study.

1. Distinction between “persistent,” “recurrent,” and “chronic” CSCR could be explained more clearly.

2. Criteria for selecting controls (e.g., refractive error, systemic diseases, medications) are not fully described. These factors may influence choroidal thickness.

3. Multiple comparisons were made without correction (e.g., Bonferroni), raising risk of type I error.

4. Lack of correlation between SFCT and PED/CST is an important negative finding, but underexplored. Discussion should consider alternative biomarkers such as choroidal vascularity index (CVI) or OCT angiography metrics.

5. Abstract and results sections use inconsistent terminology (CMT vs CST). Terminology should be standardised.

6. p-value reporting should follow uniform format (e.g., “p < 0.05” rather than “p=<0.05”).

Reviewer 2

The title of this study suggests that the analysis will look for differences in choroidal thickness and PED for acute and chronic CSCR. However the results and conclusion do not offer any differences or similarities between the two.

The significance of PED morphology was not clear.

The conclusion of increase SFCT and PED serving as a significant marker in CSCR is not new.

It is unclear which OCT slice or if more than one was used to measure the choroidal and PED measurements. Also PED measurements appear to be CST, but not always is the PED at the fovea. How would the PED be measured if its not foveal?

I have not learnt what is difference between measurements of acute or chronic changes.

Reviewers' comments:

Reviewer's Responses to Questions

**Comments to the Author**

1. Is the manuscript technically sound, and do the data support the conclusions?

Reviewer #1: Yes

Reviewer #2: Partly

2. Has the statistical analysis been performed appropriately and rigorously?

Reviewer #1: Yes

Reviewer #2: I Don't Know

3. Have the authors made all data underlying the findings in their manuscript fully available?

Reviewer #1: Yes

Reviewer #2: Yes

4. Is the manuscript presented in an intelligible fashion and written in standard English?

Reviewer #1: Yes

Reviewer #2: Yes

Reviewer #1: This study examines the relationship of subfoveal choroidal thickness (SFCT) and pigment epithelial detachment (PED) morphology with disease activity in CSCR, using OCT data from a Nepalese cohort. The manuscript provides region-specific normative SFCT values, which are lacking in current literature, and adds to the understanding of pachychoroid spectrum disorders. The findings are clinically relevant, particularly in differentiating acute from chronic disease, and in identifying biomarkers of progression. However, there are view concerns that could impact the clarity of the study.

1. Distinction between “persistent,” “recurrent,” and “chronic” CSCR could be explained more clearly.

2. Criteria for selecting controls (e.g., refractive error, systemic diseases, medications) are not fully described. These factors may influence choroidal thickness.

3. Multiple comparisons were made without correction (e.g., Bonferroni), raising risk of type I error.

4. Lack of correlation between SFCT and PED/CST is an important negative finding, but underexplored. Discussion should consider alternative biomarkers such as choroidal vascularity index (CVI) or OCT angiography metrics.

5. Abstract and results sections use inconsistent terminology (CMT vs CST). Terminology should be standardised.

6. p-value reporting should follow uniform format (e.g., “p < 0.05” rather than “p=<0.05”).

Reviewer #2: The title of this study suggests that the analysis will look for differences in choroidal thickness and PED for acute and chronic CSCR. However the results and conclusion do not offer any differences or similarities between the two.

The significance of PED morphology was not clear.

The conclusion of increase SFCT and PED serving as a significant marker in CSCR is not new.

It is unclear which OCT slice or if more than one was used to measure the choroidal and PED measurements. Also PED measurements appear to be CST, but not always is the PED at the fovea. How would the PED be measured if its not foveal?

I have not learnt what is difference between measurements of acute or chronic changes.

**Do you want your identity to be public for this peer review?** For information about this choice, including consent withdrawal, please see our Privacy Policy

Reviewer #1: No

Reviewer #2: **Yes: ** Elisa Cornish

---

## [Author Response · Author response to Decision Letter 1]

3 Oct 2025

EDITOR AND REVIEWERS’ COMMENTS AND RESPONSES

Comment 1 (Editor):

Please include this is a study of Nepalese patients in the title.

Response:

Thank you for the suggestion. The title has been revised to:

“A comparative study of choroidal thickness and pigment epithelial detachment in acute and chronic central serous chorioretinopathy in Nepalese patients”

Comment 2 (Editor):

The abbreviations and terminology related to subfoveal choroidal thickness, chronic serous chorioretinopathy and subfield choroidal thickness are not carefully defined and appear to be used interchangeably in places. Please review and please be consistent throughout.

Response:

We appreciate this observation. All abbreviations and related terminologies (e.g., CSCR, SFCT, CST) have been reviewed for consistency. We have now consistently used CST (central subfield macular thickness) as the standard terminology to denote the mean macular thickness of central subfield of 1mm in size on OCT scan. Definitions of CST and SFCT and have been clearly provided in the Methods section, and uniform terminology is now maintained throughout the manuscript.

Comment 3 (Editor):

The control patients need to be explained clearly. Also in Table 5, there is a column with non-affected fellow eyes? What are the controls specifically used for the study please.

Response:

Control subjects were age-matched individuals with no history of CSCR or retinal disease, and free from systemic diseases or medications that could influence choroidal thickness. We have further clarified the selection criteria for the control group in the Methods section. (Page 4, Line 79-85)

The “non-affected fellow eyes” in Table 5 represent eyes of unilateral CSCR patients, used as internal comparisons rather than healthy controls. As these differences had been described in previous results and objective of table 5 was not to analyse these differences, we have removed that particular column from the table 5. Thank you for pointing this out.

Comment 4 (Editor):

The distinction between acute and chronic/persistent (same thing?) disease is not established clearly in the Methods, nor is there comparative data, as noted in the title. Please clarify or change the title to reflect the exact purpose of the study.

Response:

We appreciate this comment. Although Method section defines chronic CSCR as combined set of recurrent and persistent CSCR, the use of these terms interchangeably throughout the title and text has created confusion to the reader.

We have revised the Methods section (Page 5, Line 85-91) to define chronic CSCR more explicitly and consistently used the same terminology throughout the title, results and text for better clarity. Comparative analyses between acute and chronic groups are now presented more clearly in the Results and Discussion sections.

Comment 5 (Editor):

For any measurements in Tables (for example, Table 5), the measurement units (microns) must be included and the number of decimal places biologically meaningful (not to 3 decimal places for example). Any abbreviations used in tables should also be fully explained so that the reader can follow the information easily.

Response:

We have added measurement units (micrometer/µm) to all relevant tables, rounded numerical values to two decimal places, and ensured all abbreviations are explained in table footnotes.

Comment 6 (Editor):

Table 4 is described as showing associations between the different pathologies, however, no statistics are reported (as far as can tell) and this seems to be a summary table of numbers of patient eyes for the different conditions. Please either analyse for associations as noted or remove this table. The same comments apply to Table 5, which also notes associations in the title but does not provide any analysis.

Response:

The associations are analysed for both tables 4 and 5 and the statistical analysis has been explained in the text part of the results section. As the association part (statistical analysis) needed some explanation, we previously chose to explain it in the text and omitted it from the tables per se to avoid duplication.

As per your suggestion, now, we have included the important results of the statistical analysis with the tables 4 and 5. (Page 10, Line: 167; Page 11, Line 183,184)________________________________________

Comment 7 (Editor):

The observations regarding PED, subfoveal choroidal thickness as biomarkers for CSCR (acute or chronic?) are reported previously in literature, and the conclusions around this finding and need for more choroidal biomarkers to be examined requires a more defined and focused statement, supported by the observations.

Response:

We acknowledge the prior literature and have revised the Discussion and Conclusion sections to reflect on this suggestion.

While our findings are not novel per se, our data contribute additional insights specific to the Nepalese population. We have added discussion about the usefulness and limitations of various markers analysed in our study and on the need for evaluation of additional biomarkers, such as choroidal vascularity index (CVI) and others. The discussion has been based on our observations and emerging new findings in CSCR. (Page 15-16: 259-277)

Comment 8 (Editor):

The literature cited is limited and not current for this area of ophthalmology... For example, Khan and Lotery, Ann Rev Vis Sci 2024...

Response:

Thank you for pointing this out. We have updated our literature review and added several recent and relevant studies, including the suggested article by Khan and Lotery (2024), and other recent studies on pachychoroid spectrum disorders and OCT biomarkers.

Newer studies have been explored and cited (Page 19, References 26-31)

Comment 9 (Editor):

With regard to limitations, please note the limited (around 140) patient numbers, and be careful using the term 'population' in referring to the study group, noting sample size.

Response:

We agree and have revised the Limitations accordingly to reflect the limited sample size. We now use the term sample, study group or subset of Nepalese population instead of "population" and have acknowledged the relatively modest sample size as a limitation that may affect generalizability.

Reviewer 1 Comments and Responses

Reviewer 1 – Comment 1:

Distinction between “persistent,” “recurrent,” and “chronic” CSCR could be explained more clearly.

Response:

Methods section provides the definitions for acute, persistent, and recurrent CSCR based on classification given by Daurich et al. with appropriate citation. Chronic CSCR has been used to include persistent and recurrent CSCR in our study and that has been clearly mentioned in the methods section. (Page 5, Lines: 85-91)

Reviewer1– Comment 2:

Criteria for selecting controls (e.g., refractive error, systemic diseases, medications) are not fully described.

Response:

The control selection criteria have been clarified in the Methods section. Controls were age- and sex-matched, with refractive error between ±1.0 D, no known systemic and ocular diseases, and no history of medications affecting choroidal thickness. (Page 4: Lines 78-84)

Reviewer 1 – Comment 3:

Multiple comparisons were made without correction (e.g., Bonferroni), raising risk of type I error.

Response:

Thank you for this valid suggestion. We acknowledge that applying the Bonferroni correction enhances result reliability by reducing familywise error, particularly in relation to our primary study objectives. Hence, we have now pre-specified three primary hypotheses addressing SFCT–PED association and the relationships of PED and SFCT with acute vs chronic CSCR, and have applied Bonferroni correction (α = 0.05/3 = 0.0167) to those tests; other analyses are presented as exploratory. This change has been reflected in the Methods section (Page 7: 131-137) and some results have been reinterpreted using this correction. (Page 10: 175-179)

Reviewer 1 – Comment 4:

Lack of correlation between SFCT and PED/CST is an important negative finding, but underexplored. Discussion should consider alternative biomarkers...

Response:

Thank you for this helpful suggestion. We have expanded the Discussion to explore this negative finding further and incorporated literature on alternative biomarkers such as CVI and other parameters that may correlate with pathophysiological changes in CSCR. (Page 15-16: 259-277)

Reviewer 1 – Comment 5:

Abstract and results sections use inconsistent terminology (CMT vs CST).

Response:

The terminology has been standardized across all sections to Central Subfield Macular Thickness (CST), in alignment with standard nomenclature.

Reviewer 1 – Comment 6:

p-value reporting should follow uniform format (e.g., “p < 0.05”).

Response:

All p-values have been revised to follow a consistent format (e.g., p < 0.05).

Reviewer 2 Comments and Responses

Reviewer 2 – Comment 1:

The title of this study suggests that the analysis will look for differences in choroidal thickness and PED for acute and chronic CSCR. However the results and conclusion do not offer any differences or similarities between the two.

Response:

We appreciate this comment as we realised that inconsistent alternate use of persistent and recurrent CSCR for chronic CSCR has created this confusion. We have defined the chronic CSCR in Methods section and consistently used chronic CSCR in results and discussion section. (Page 5: Lines: 85-91)

Comparative results between acute and chronic CSCR are now more clearly emphasized in the Results section in Table 5 and its preceding text. (Page: 10, Lines: 169-179)

Reviewer 2 – Comment 2:

The significance of PED morphology was not clear.

Response:

The role of PED morphology has been explored in both the Results and Discussion, with emphasis on its distribution between acute and chronic CSCR and potential clinical implications. The results and discussion section also include the analysis and implication of those analysis as detailed below:

Results: Although there was no significant difference in presence of isolated flat PEDs between acute and chronic CSCR eyes, we found that overall flat PEDs (present solitarily or mixed with dome shaped PEDs) were significantly more common in chronic CSCR eyes (p=0.027). (Page 10: 169-174)

Discussion: Some studies have reported the presence of fibrovascular components in flat and irregular PEDs.[23,24] Bosquet et al. found a fibrovascular component in as many as one-third of their patients with flat and irregular PEDs. In their study, only flat PEDs with hyperreflective contents were associated with neovasculopathy. In our study, we did not assess the content of all flat PEDs with angiography. However, a significant number of these flat PEDs were observed even in the early course of the disease, and many were also present concurrently with dome-shaped PEDs. Hence, we anticipate that a substantial proportion of these PEDs may not be associated with neovascular changes. Given the reports of fibrovascular components in these flat PEDs, it may be logical to evaluate them using optical coherence tomography angiography (OCTA) or conventional angiography, especially if they exhibit hyperreflective content and the disease is persistent or recurrent. Our study also demonstrated a significantly higher number of flat PEDs in chronic and recurrent cases, similar to findings reported in some other studies. [9,22] (Page 14: 229-241)

Reviewer 2 – Comment 3:

The conclusion of increased SFCT and PED as markers in CSCR is not new.

Response:

We acknowledge this fact. While the markers are established, our study contributes population-specific normative values and reinforces the applicability of these markers in a South Asian cohort.

In addition, our study also explores the variations of these markers between acute and chronic CSCR eyes and to study their correlation to each other.

Reviewer 2 – Comment 4:

It is unclear which OCT slice or if more than one was used... How was PED measured if not foveal?

Response:

The PED sizes were not analysed in our study. We evaluated presence or absence of PED and the PED morphology was classified if present. Any PED present within the central 9 by 9 mm OCT scan was analysed.

As rightly pointed out, we had failed to clarify the OCT sections used for this analysis in our methods section. The Methods section, with correction, now specifies that presence of PED and its morphology was recorded based on review of complete 9 by 9 mm 24-line volume OCT scan. (Page 6: 115-116)________________________________________

Reviewer 2 – Comment 5:

I have not learnt what is the difference between measurements of acute or chronic changes.

Response:

We realized that there was some confusion regarding comparison between acute and chronic CSCR due to inconsistent use of terminologies in the result section. We have restructured the Results section to clearly delineate the comparative measurements of CST, SFCT and PED between acute and chronic CSCR groups, with accompanying statistical analysis. (Result section: Table 5 and preceding text Page: 10, Lines: 169-179)

---

## [Decision Letter · Decision Letter 1]

19 Oct 2025

A comparative study of choroidal thickness and pigment epithelial detachment in acute and chronic central serous chorioretinopathy in Nepalese patients

PONE-D-25-32356R1

Dear Dr. Bhatta

We’re pleased to inform you that your manuscript has been judged scientifically suitable for publication and will be formally accepted for publication once it meets all outstanding technical requirements.

Kind regards,

Jiro Kogo

Academic Editor

PLOS ONE

Additional Editor Comments (optional):

Reviewers' comments:

Reviewer's Responses to Questions

**Comments to the Author**

Reviewer #1: All comments have been addressed

Reviewer #2: All comments have been addressed

2. Is the manuscript technically sound, and do the data support the conclusions?

Reviewer #1: Yes

Reviewer #2: Yes

3. Has the statistical analysis been performed appropriately and rigorously?

Reviewer #1: Yes

Reviewer #2: I Don't Know

4. Have the authors made all data underlying the findings in their manuscript fully available?

Reviewer #1: Yes

Reviewer #2: Yes

5. Is the manuscript presented in an intelligible fashion and written in standard English?

Reviewer #1: Yes

Reviewer #2: Yes

Reviewer #1: All the comments were addressed and can go ahead further with the process.

Reviewer #2: (No Response)

**Do you want your identity to be public for this peer review?** For information about this choice, including consent withdrawal, please see our Privacy Policy

Reviewer #1: No

Reviewer #2: No

---

## [Editor Report · Acceptance letter]

PONE-D-25-32356R1

PLOS ONE

Dear Dr. Bhatta,

I'm pleased to inform you that your manuscript has been deemed suitable for publication in PLOS ONE. Congratulations! Your manuscript is now being handed over to our production team.

Kind regards,

on behalf of

Prof. Jiro Kogo

Academic Editor

PLOS ONE